# Academic Performance and Peer or Parental Tobacco Use among Non-Smoking Adolescents: Influence of Smoking Interactions on Intention to Smoke

**DOI:** 10.3390/ijerph20021048

**Published:** 2023-01-06

**Authors:** Angdi Zhou, Xinru Li, Yiwen Song, Bingqin Hu, Yitong Chen, Peiyao Cui, Jinghua Li

**Affiliations:** School of Public Health, Jilin University, Changchun 130021, China

**Keywords:** intention to smoke, peer smoking, parental smoking, academic performance, adolescence

## Abstract

Background: Intention to smoke is an important predictor of future smoking among adolescents. The purpose of our study was to examine the interaction between academic performance and parents/peer tobacco use on adolescents’ intention to smoke. Methods: A multi-stage stratified sampling was used to select participants, involving 9394 students aged between 9–16 years in Changchun city, northeastern China. Multiple logistic regression analyses were conducted to examine the individual effect of academic performance and peer/parental smoking behavior. Stratified logistic regressions were conducted to examine the protective effect of academic performance based on peer or parental smoking. Interaction effects of academic performance × peer/parental smoking on adolescents’ intention to smoke were tested. Results: Of all the non-smoking students sampled, 11.9% intended to smoke within the next five years. The individual effect of academic performance and peer/parental smoking was significant. The protective effect of academic performance on the intention to smoke was significant regardless of whether peers smoked or not. However, the protective effect was not significant among adolescents with only maternal smoking and both parental smoking. The current study found the significant interaction effects of academic performance × peer smoking and the academic performance × both parents’ smoking. Students with poor academic performance were more likely to intend to smoke if their peers or both parents smoked. Conclusion: These preliminary results suggest that peer smoking or smoking by both parents reinforces the association between low academic performance and the intention to smoke among adolescents. Enhancing school engagement, focusing on social interaction among adolescents with low academic performance, and building smoke-free families may reduce adolescents’ intention to smoke.

## 1. Introduction

Tobacco use by adolescents is a global public health concern. Surveys of tobacco use among adolescents aged between 13–15 in Asia reported that the current cigarette smoking prevalence among adolescents was 3.9% in China, 3.4% in Iran, and 4.3% in Singapore [1]. However, the adult smoking rate in China was as high as 26.6% [2] compared with 12% in Iran [3] and 16% in Singapore [4]. As smoking prevalence among adolescents is low but adult smoking prevalence is high, the driving factors deserve more attention. Behavioral exposures resulting from high social smoking rates in Chinese society may be one of the factors that motivate non-smoking adolescents to become smokers as adults [5]. That means that non-smoking adolescents are influenced by smokers in their families, schools, and communities to gradually become smokers [6,7]. Specifically, the process would be long, and adolescents may not decide to smoke immediately due to practical constraints but, rather, may try to smoke in the future after developing an intention to smoke.

Intention to smoke is a valid and reliable predictor of smoking behavior [8]. According to the theory of planned behavior, intentions and behaviors are consistent [9]. Specifically, social behavior is a goal-oriented outcome and a direct result of the intention [10]. Although there is a gap between the number of adolescents who intend to smoke and the prevalence of smoking in matched ages [11], the intention to smoke among adolescents is closely linked to adult smoking, which has been examined in a variety of studies [8,12,13,14]. Several longitudinal studies conducted in Europe have shown that the onset of tobacco use in numerous lifelong smokers can be traced back to their cognition and attitude toward tobacco use in middle school or even the elementary school [15,16]. Thus, adolescence is a prime period for tobacco intervention—it is an urgent public health goal to identify the factors influencing smoking intentions among the youth.

Prior studies have placed special emphasis on the impact of the school environment on children’s behaviors and attitudes [17,18] as school is the primary environment in which adolescents grow up and acquire necessary knowledge and skills. The social-control theory asserts that strong social bonds, which consist of four main factors (attachment, commitment, involvement, and belief), can inhibit individuals from engaging in risky behaviors [19]. Individuals are not bound by norms if they are not concerned with their social connections to others [20]. The stronger the bond, the less likely it is that adolescents tend to deviate [21]. Youth achievement in school usually represents social bonding elements in the social-control theory [22]. High-academic achievers who are recognized by their teachers, peers, and parents typically establish strong school ties, engage in the school environment [23], and establish psychological identification with their internal school rules [24], thus avoiding the intention to smoke. Low academic achievers are commonly not recognized in the campus environment, which results in a weaker connection to their campus and a reluctance to follow campus rules [25]. Therefore, adolescents’ weak ties with the campus due to low academic performance result in an increased risk of smoking interest in adolescence [26,27]. In support of this view, previous studies have noted that academic performance and intentions to smoke are highly correlated during adolescence [27,28,29,30]. A large-scale longitudinal study in Finland showed that students with poor academic performance were six times more likely to smoke regularly than those with good academic performance [31]. In China, evidence from a multiethnic survey found that students with academic performance in the last 25% and lower than 50% categories were more likely to have the intention to smoke [32]. Moreover, academic performance was shown to reflect students’ ability to control or manage themselves [33]. Researchers further pointed out that lower achievers may be more vulnerable to smoking because they feel less capable of resisting various temptations [25,28]. Based on the above literature, the relationship between adolescents’ academic performance and other smoking risk factors may vary at different academic performance levels. If so, youth smoking prevention programs may benefit from this knowledge and can potentially be improved.

According to the social learning theory [34,35] smoking intentions and behavior can be acquired through a learning process in which imitation and differential reinforcement play key roles [36,37]. This indicates that the perceptions and behaviors of smoking can be affected by environmental exposures, such as intergenerational influences in the family and peer influences from shared settings [38,39]. Parental smoking behavior may serve as a model for adolescents and be interpreted by their children as a tacit license for addictive-substance use [40]. Prior research discovered that adolescents whose parents smoke are more likely to smoke in the future than children in tobacco-free homes [41]. Moreover, a large body of published research supports the idea that their close friends’ smoking increases the risk of future smoking among adolescents [42], and young people’s initiation of smoking is likely triggered by their best friends [43]. In-school groups are a traceable starting point for the developmental trajectory of adolescents’ substance-use intentions [44,45]. Once someone in the group smokes, those involved in tobacco use force the negative norms and values on the group members, thereby changing non-smokers’ perceptions about tobacco to be consistent with the expectations of the group [46].

Bronfenbrenner’s human ecological perspectives provide a conceptual framework for our study of the interaction. Bronfenbrenner argued that human development occurs in a complex set of nested environments [47]. Family, peer, and school environments are defined as microsystems that belong to the proximal environment and have a specific and direct influence on adolescents’ behavioral intentions. Although early studies suggest that adolescent tobacco-use intentions and behaviors among adolescents are directly influenced by microsystems such as school involvement and family environment [48,49], there is limited research on the combined effects of these proximal systems. Research on adolescent smoking intentions needs to be expanded to mesosystems that include interactions between families, peer groups, and schools and receive more attention. Poor academic performance and parental/peer smoking may coexist; however, few studies have considered the interaction effect on adolescent smoking intentions. Compared with the intervention of a single risk factor, exploring the combined effect of multiple risk factors will help formulate adolescent health-promotion strategies and prevent the occurrence of smoking intentions and behaviors in multiple contexts [50]. Therefore, this study aimed to explore the interaction between academic performance and parental/peer smoking on adolescents’ intention to smoke. Our study will provide a more thorough understanding of the link between smoking risk factors and a better understanding of the smoking process and potentially identify more intervention targets.

## 2. Materials and Methods

### 2.1. Data Collection and Participants

Data from this study were collected as part of a cross-sectional study on the prevalence of health risk behaviors in primary and secondary school students in Changchun, Jilin Province, China. Changchun is located in northeastern China, has a population of over nine million, and is a core city in the high-latitude region of China. The 2013 China Urban Tobacco Prevalence Survey, organized by the Chinese CDC and the US CDC, showed that the overall smoking rate among urban adults (aged 15+) in Changchun was 23.5%, with 43.2% for men and 3.8% for women [51], which is similar to national level. Thus, Changchun is a typical city for studying tobacco use among adolescents.

This study was conducted in collaboration with Changchun Center for Health Education. The samples for this study were selected using multistage stratified cluster sampling. In the first stage, six districts with different economic levels in Changchun city were selected. Second, three secondary schools and three elementary schools in each district were randomly chosen. Survey participants comprised students in grades 4–6 in elementary schools and grades 7–8 in middle schools. Third, whole-group sampling was conducted, using 2–3 classes in each grade for each target school. Participants who refused to answer were excluded, and 9893 questionnaires were collected after distributing 10,157 questionnaires, with a response rate of 97.4%. After removing the samples with smoking experience (N = 227), a missing dependent variable (intention to smoke) (N = 255), or having logical contradictions (N = 17) in their answers, 9394 participants were selected.

The survey was a self-administered questionnaire provided to the students by teachers and school physicians. Teachers and school physicians were trained by the Changchun Center for Health Education. Teachers and school physicians could only respond to the questions asked by the students and did not interfere with their responses. To screen out smokers, all respondents completed questions about their smoking experiences and current smoking. Former and current smokers were excluded from this study. Permission to carry out the survey was obtained from students and their guardians. In the present study, a passive-informed consent procedure was used in which adolescents and their guardians were informed about the study by the school before data collection. During the school day, data collection was conducted in the classroom, and the students answered the questionnaire anonymously. If students or their guardians refused to participate in the study, they signed an informed consent form at the school or provided a verbal notification before the start of the survey. The study was approved by the Medical Ethics Committee of the School of Public Health at Jilin University. The approval number is 2020-10-16.

### 2.2. Measures

#### 2.2.1. Smoking Experience 

The students were asked to indicate their smoking experience using two questions: (1) “Have you ever smoked (even one or two puffs; smoking in this questionnaire includes cigarettes and other tobacco products such as e-cigarettes, hookahs, etc.)”? (2) “Have you smoked in the past 30 days”? The response options were “yes” and “no”. Participants who answered “no” to both questions were considered non-smokers.

#### 2.2.2. Academic Performance

Students were asked to indicate their average school-grade ranking in their class for the most recent academic year. The response options were as follows: 1 = poorest, 2 = lower 20%, 3 = medium 20%, 4 = higher 20%, and 5 = excellent. 

#### 2.2.3. Parental Tobacco Use

The students were asked to indicate their parental tobacco use by answering one question: (1) “Does any of the following people smoke at home”? The response options were as follows: 1 = none, 2 = only mother, 3 = only father, and 4 = both father and mother.

#### 2.2.4. Intention to Smoke

Students were asked to indicate their intention to smoke by answering the following question: (1) “Do you think you will smoke within the next five years”? The response options were 1 = definitely no, 2 = probably no, 3 = probably yes, and 4 = definitely yes. Based on a rationale derived from Pierce et al.’s method of assessing smoking susceptibility [48,52,53], those who answered “probably no”, “probably yes”, and “definitely yes” were classified as “intend to smoke”, and those who answered, “definitely no” were classified as ”don’t intend to smoke”.

#### 2.2.5. Peer Tobacco Use

The students were asked to indicate their peers’ tobacco use using one question: (1) “How many of your friends smoke”? Responses were rated on a 4-point Likert-type scale: 1 = none, 2 = some, 3 = most, and 4 = all. Responses were recorded as follows: 1 = no peers smoking; 2 = peer smoking.

#### 2.2.6. Control Variables

Other variables that could confound the relationship between parental or peer tobacco use, academic performance, and adolescents’ intention to smoke included sex, age, parental education level, family structure, and self-efficacy. These variables were selected as control variables because they were significantly associated with adolescents’ intentions to smoke [54,55,56]. The information about the highest level of education received by the parents was obtained via two questions: (1) “What is your mother’s highest attained education”?(2) “What is your father’s highest attained education”? In our study, we divided the answers into three options (1 = middle school and below; 2 = high school; 3 = undergraduate and above). 

This study used the General Self-Efficacy Scale to measure the self-efficacy of non-smoking elementary and middle school students. This scale was introduced and applied in China, and it has been proven to have good reliability and validity [57]. In our study, the KMO value of the General Self-Efficacy Scale is 0.901(>0.85). Bartlett’s spherical test showed a significant statistical result (*p* < 0.001). The Cronbach’s alpha of the scale was 0.845. These results indicated a good construct validity and high reliability of the General Self-Efficacy Scale in this survey. This is a one-dimensional scale with a total of 10 items scored on a 4-level scale. The answers were: 1 = completely incorrect, 2 = somewhat correct, 3 = mostly correct, and 4 = completely correct. The total score was 40 points. The higher the score, the higher was the general self-efficacy. To avoid interference from different family compositions, the family structure of adolescents was also investigated and used as a control variable in this study. Adolescents were asked to complete a questionnaire about the roster of family members living together over time, which we used to construct a report measuring adolescents’ perceptions of family structure. It was coded into three types of family structure: (a) nuclear family: families with two biological married parents, (b) stem family: families with two biological married parents and grandparents; (c) other families: including single-parent families, reconstituted families, and types of families not mentioned above.

### 2.3. Data Analysis

All data were analyzed using R 4.0.3. Frequencies were calculated to summarize the distribution of categorical variables. The chi-square test was used to test the distribution of the risk factors for future smoking intentions. Among non-smoking adolescents, a binary logistic model was developed using parental smoking, peer smoking, and self-rated academic performance as independent variables and intention to smoke (not intended to smoke = 0/intend to smoke = 1) as the dependent variable. Subsequently, stratified models were developed to examine the associations between academic performance, parental/peer smoking, and intention to smoke among adolescents. Finally, to explore the interaction between academic performance and parental/peer smoking behavior on intention to smoke, several logistic models were developed with academic performance × tobacco use by peers/parents as the interaction term. Interaction terms were created by multiplying the peer smoking or parental smoking variables and academic performance. All the above models used sex, age, parental education level, family structure, and self-efficacy as control variables. Continuous variables were centered, while categorical variables were treated as dummy variables. In the regressions, the missing values of the independent variables were replaced by multiple imputations. Statistical tests were performed using a two-sided test, with *p* < 0.05 indicating a statistically significant difference. 

## 3. Results

### 3.1. Participants’ Characteristics

Overall, 9394 non-smoking students aged between 9–16 participated in the study. The mean age (and SD) of the non-smoking students was 12.32 (1.5) years. Of these, 8273 (88%) of the non-smoking students reported definitely not smoking within the next five years, 797 (8.5%) reported probably not, 288 (3.1%) reported probably yes, and 36 (0.4%) reported definitely yes. 

Of the participants, 14.97% of boys intended to smoke in the next five years, which is more than girls (χ^2^ = 85.1, *p* < 0.001); by age, a higher proportion of adolescents aged 15 or older intended to smoke (χ^2^ = 85.1, *p* < 0.001); by parental education level, a higher proportion of adolescents whose mothers (χ^2^ = 23,669, *p* < 0.001) or fathers (χ^2^ = 28,021, *p* < 0.001) with an education level of junior high school and below intended to smoke; by family structure, a higher proportion of adolescents from other families intended to smoke (χ^2^ = 39,118, *p* < 0.001). The prevalence of intention to smoke was higher among those with low general self-efficacy (χ^2^ = 100,525, *p* < 0.001) and low academic performance (χ^2^ = 158,305, *p* < 0.001). A total of 28.76% of adolescents who reported peer smoking intended to smoke in the next five years, which was more than adolescents whose peers did not smoke(χ^2^ = 376,446, *p* < 0.001). By parental smoking, a higher proportion of adolescents whose both parents smoked intended to smoke (χ^2^ = 111,128, *p* < 0.001). (Please see Table 1).

### 3.2. Risk Factors for Adolescents’ Intention to Smoke

As shown in Table 2, after controlling for sex, age, family structure, mother’s and father’s education level, and general self-efficacy, the logistic regression analysis revealed that adolescents with good academic performance (OR = 0.80, 95% CI = 0.75–0.85) were less likely to smoke in the future. Compared to adolescents without peer smoking, those whose peers smoked (OR = 2.92, 95% CI = 2.49–3.43) were more likely to have the intention to smoke. Compared to non-parentally smoking students, students in smoking families were more likely to have the intention to smoke: (1) only father smoking (OR = 1.51, 95% CI = 1.31–1.74) and (2) both parents smoking (OR = 2.31, 95% CI = 1.77–3.02). In addition, sex (ref = girls), age, family structure (ref = nuclear family), and general self-efficacy were significantly associated with the intention to smoke.

As shown in Table 3, in stratified models, for students without smoking peers, the OR of academic performance was 0.83 (0.78–0.89); with smoking peers, the OR of academic performance was 0.76 (0.73–0.79). Without smoking parents, the OR of academic performance was 0.72 (0.67–0.76); with only the father smoking, the OR of academic performance was 0.80 (0.76–0.84). For those whose mothers smoked or both parents smoked, the effect on academic performance was not significant.

Logistic regression models were also tested for interactions between academic performance and parental/peer smoking on smoking intentions. The results showed significant interaction effects of academic performance × peer smoking (OR = 1.21, 95% CI = 1.05–1.38) and academic performance × both parents’ smoking (OR = 1.39, 95% CI = 1.10–1.76). The results of these separate logistic regression models are summarized in Table 4.

### 3.3. Academic Performance and Peer Smoking Interactions on the Intention to Smoke among Non-Smokers

The interaction between academic performance and peer smoking was positive and significant (*p* < 0.01). Figure 1 shows the relationship between academic performance and intention to smoke for respondents whose peers smoked and those whose peers did not. The graph shows that individuals with a lower academic performance whose peers smoked had a higher intention to smoke. Whether or not their peers smoked, their intention to smoke declined as academic performance increased. 

### 3.4. Academic Performance and Both Parents’ Smoking Interactions on the Intention to Smoke among Non-Smokers

The interaction term between academic performance and both parents’ smoking was positive and significant (*p* < 0.01). Figure 2 shows the relationship between academic performance and intention to smoke for respondents whose parents smoked. Specifically, low-academic achievers whose parents smoked had significantly higher intentions to smoke than those whose parents did not smoke and who demonstrated excellent academic performance. For those with non-smoking parents, intentions to smoke declined as academic performance improved. In contrast, for those students with both parents smoking, smoking intentions were already high and did not change significantly as academic performance improved. Thus, we verified this hypothesis.

## 4. Discussion

In this study, the individual effects of academic performance, peer and family members’ tobacco use, and significant interactions on adolescents’ intention to smoke were found. The results showed that peer smoking and both parents’ smoking reinforced the risk of future smoking among low-academic achievers. These findings may help us to understand the combined impact of academic difficulties and the demonstration of parental/peer smoking, and explore the role of family, peer, and school environment in the development of adolescent smoking intentions in a comprehensive manner, emphasizing the value of this joint contextual perspective in adolescent smoking-prevention research. There are potential pathways to high-smoking risk that deserves more attention than focusing on individual factors. Furthermore, these findings will hopefully have implications for policymakers in youth smoking prevention.

In the current study, 11.9% of non-smoking adolescents aged between 9–16 years in Changchun reported being likely to smoke within five years, which was slightly higher than the 9.7% of Chinese middle school students aged between 13–15 in another study conducted in 2015 [32]. Previous research supported that adolescents eventually becoming avid smokers appears to be the result of having developed cognitive susceptibility to smoking [48]. Adolescence serves as a “readiness period”, in which adolescents form expectations and beliefs about smoking. Therefore, government departments and schools must take relevant measures to reduce teenagers’ intention to smoke and prohibit smoking. Strict smoking bans in schools are thought to have contributed to the decline in smoking intention and current smoking rates. In recent years, globally, school-smoking ban policies have been implemented [58]. Currently, smoking on school grounds is explicitly prohibited for students in China, but there are limited restrictions on teachers smoking on campus. Teacher-smoking at school is tolerated by some students and their families [59]. The teacher is undoubtedly a potential professional that can act as a smoking role model for children because of their position of authority and frequent contact with the students [60]. Therefore, it is urgent to carry out a strict smoking ban for the construction of smoke-free campuses in China, not only for school students but also for all campus staff. Health education in school is also an important part of tobacco prevention and control for adolescents; thus, health education on tobacco must be included in the school curriculum [61,62].

The results of the stratified regressions indicated that the high academic performance of students was negatively related to the intention to smoke regardless of whether peers smoked. This protective effect was also observed in participants with non-smoking parents or only fathers who smoked. Academic performance is an important indicator of the success of a campus. Academic performance is usually the only criterion for evaluating student achievement, especially in the Chinese social context. Academic performance is not only related to adolescent’s sense of accomplishment but also to the social and emotional support they received [63]. According to the social control theory, adolescents with good academic performance tend to have a strong connection with schools and are more inclined to abide by school rules and reduce their smoking intention. However, academic performance is not the only factor that can strengthen school bonds [21]. Schools should use diverse indicators to evaluate students’ achievements rather than a single evaluation indicator, such as academic performance. Numerous activities should be organized for different students to allow them to realize their own value and strengthen their bonds with the school. This will facilitate compliance with campus norms for low-academic achievers and reduce their intention to smoke [26].

Tests of interaction showed that adolescents who did poorly in school and whose peers smoked were more likely to smoke within the next five years than those who did well in school or whose peers did not smoke. From the social-control theory perspective, academic performance usually represents social bonding among adolescents, and weak school bonds can predict a low adherence to school norms and poor self-control capability. Therefore, low-academic students are more likely to be influenced by peer modeling. While few studies have examined interactions between school ties and peer smoking, our findings are consistent with adolescent behavior development research, suggesting that the joint effect of weak school bonds and peer influence could reinforce the risk of youth misbehavior [64,65]. The results suggest that more attention should be paid to the transmission of smoking behavior among groups of adolescents with low academic performance. Schools and parents need to care about the socialization of low achievers and encourage them to participate in school activities to build confidence [66]. Smoking-free policies should be maintained on campus and in the community to prevent youth from accessing tobacco products and creating an epidemic within youth groups [67]. Peer pressure and modeling can also be used to control tobacco use among minors [68]. Specifically, health education in schools can invite former smokers who quit smoking to share their experiences with quitting to change misconceptions about smoking among adolescents who intend to smoke. Building smoke-free schools can also reshape peer norms, thereby reducing members’ intentions to smoke and resisting smoking when in groups [69].

In this study, adolescents whose parents smoked had a higher intention to smoke within the next five years than those whose parents did not smoke, regardless of their level of academic performance. In addition, for those whose parents did not smoke, their intention to smoke decreased as academic performance increased. Thus, the differences in the effects of parental smoking became more pronounced as academic performance increased. However, it is noteworthy that we did not find significant interactions between academic performance and only father/only mother smoking behaviors on their children’s smoking intention, which may imply that these factors individually affect smoking intentions rather than in combination with academic performance. This specific finding supported the theory that dual-smoking role models within the family increase the risk of adolescent smoking, not only exerting a stronger influence in comparison to a single role model but also weakening the protective effect of academic performance on adolescent smoking intention. This may be attributed to the fact that families in which both parents smoke prevent adolescents from having a smoke-free role model [70]. Adolescents with two smoking parents tend to lack effective smoke-free guidance in their families and are also exposed to secondhand smoke, which may lead to nicotine addiction and increase their willingness to smoke [71]. Therefore, building smoke-free homes will not only prevent adolescents from directly imitating their parents’ smoking behaviors but also protect them from the dangers of secondhand smoke.

Our findings support Bronfenbrenner’s ecology from a human-development perspective. Specifically, adolescents live in a layered and complex social environment, and their intention to smoke is influenced by their interactions with proximal environments. Therefore, studies focusing on a single environment are likely to overestimate their contribution to adolescents’ intentions to smoke. However, the interaction between multiple environments may weaken or amplify specific effects. This study may provide an empirical basis for highlighting the important role of family/peer modeling and school bonds: prevention of adolescent smoking intentions requires comprehensive interventions based on proximal environments, taking into account school, peers, and family. Overall, holistic programs derived from real-life interactions should be considered in guiding the development of youth attitudes toward misbehaviors [72]. Continued examination of the interaction effect between environmental factors and social bonds is critical to better understanding the mechanisms underlying the formation of the intention to smoke among non-smoking students.

## 5. Limitations

This study has several limitations. First, given the cross-sectional design of this study, causality cannot be inferred from the identified associations. However, the association between the identified correlates and future intention to smoke in this study existed before adolescent smoking, thus providing information for subsequent studies. Second, considering that the questionnaire was self-administrated and answers were not biologically validated, participants may have withheld some information in the survey conducted in schools, such as peer smoking and their own intention to smoke. Third, although the influences of school and family are effective over time [73], adolescence is a special period with rapid cognitive development. Many underlying psychological characteristics and attitudes toward addictive substances may be age-related. However, the current investigation has limited coverage of the relevant content. In addition, as numerous tobacco control policies are implemented and health education becomes more widespread, adolescents’ increasing age and cognitive function may lead them to resist tobacco exposure at home, school, and in the community. Future research can attempt to identify and explain age-related changes in adolescent samples. Fourth, the sample for this study came from the same city. Therefore, cultural and geographical differences were ignored. Future research should also extend to a comparison of adolescents’ smoking intentions in different cultural contexts and consider investigating the effects of local campus policies and health education on adolescents’ smoking perceptions and behaviors.

## 6. Conclusions

In the current study, the effects of positive interaction between academic performance and peer smoking and between academic achievement and parental smoking were found in the process of forming intentions before adolescent smoking. These findings emphasize that government health and education agencies and local schools should focus on student–school connections and peer interactions for adolescents with low academic performance. Parents should make every effort to avoid smoking in front of their children, which may increase the risk of future smoking among adolescents. Public health programs, education departments, and communities should incorporate smoke-free environments into the daily lives of adolescents, especially in school and home environments.

## Figures and Tables

**Figure 1 ijerph-20-01048-f001:**
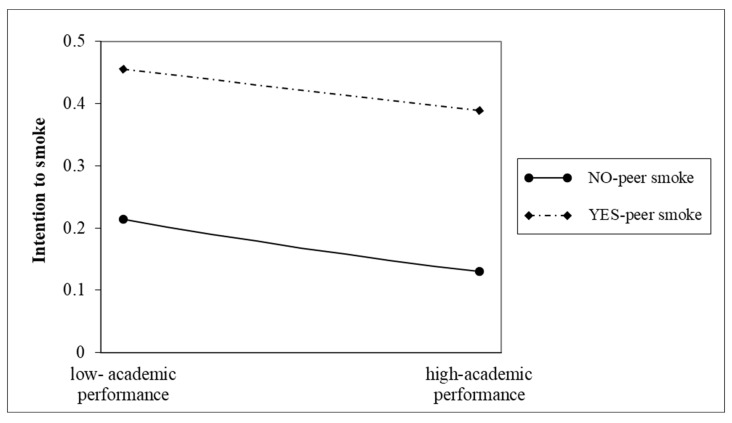
Interaction between academic performance and peer smoke and smoking intentions among non-smoking students.

**Figure 2 ijerph-20-01048-f002:**
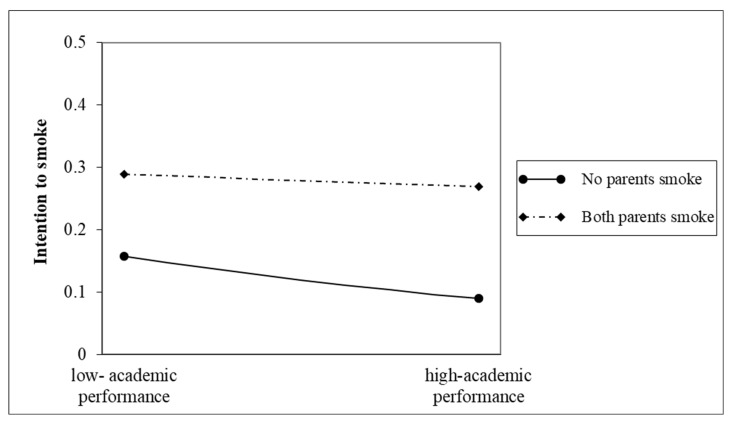
Interaction between academic performance and parents’ smoking and intention to smoke among non-smoking students.

**Table 1 ijerph-20-01048-t001:** Participant characteristics among students who are non-smokers.

Variables		Not Intend to Smoke	Intend to Smoke	Total	χ^2^	*p*-Value
		n (%)	n (%)	n		
Sex	girls	4218 (91.20)	407 (8.80)	4625	85,100	<0.001
	boys	4055 (85.03)	714 (14.97)	4769		
Age	~10	1018 (92.63)	81 (7.37)	1099	152,448	<0.001
	~12	3505 (91.37)	331 (8.63)	3836		
	~14	3248 (85.32)	559 (14.68)	3807		
	≥15	362 (76.37)	112 (23.63)	474		
Mother’s education level	middle school and below	3953 (87.32)	574 (12.68)	4527	23,669	<0.001
	high school	3053 (87.65)	430 (12.35)	3483		
	undergraduate and above	1234 (92.09)	106 (7.91)	1340		
Father’s education level	middle school and below	3770 (86.81)	573 (13.19)	4343	28,021	<0.001
	high school	3167 (88.27)	421 (11.73)	3588		
	undergraduate and above	1296 (92.05)	112 (7.95)	1408		
Family structure	nuclear family	4292 (88.35)	566 (11.65)	4858	39,118	<0.001
	stem family	2058 (90.70)	211 (9.30)	2269		
	other family	1903 (84.73)	343 (15.27)	2246		
General self-efficacy	<20	1333 (82.44)	284 (17.56)	1617	100,525	<0.001
	20–30	4801 (87.72)	672 (12.28)	5473		
	>30	2133 (92.90)	163 (7.10)	2296		
Academic performance	poorest	546 (79.13)	144 (20.87)	690	158,305	<0.001
	lower 20%	1552 (83.08)	316 (16.92)	1868		
	middle 20%	3193 (88.79)	403 (11.21)	3596		
	higher 20%	2247 (91.08)	220 (8.92)	2467		
	excellent	711 (95.44)	34 (4.56)	745		
Peer smoking	no	7359 (90.63)	761 (9.37)	8120	376,446	<0.001
	yes	857 (71.24)	346 (28.76)	1203		
Parental smoking	no parents smoke	4073 (91.08)	399 (8.92)	4472	111,128	<0.001
	only mother smoke	114 (80.85)	27 (19.15)	141		
	only father smoke	3712 (86.37)	586 (13.63)	4298		
	both parents smoke	341 (76.80)	103 (23.20)	444		

**Table 2 ijerph-20-01048-t002:** Multivariate logistic model for factors associated with intention to smoke among non-smoking students.

Variables	Estimate	Std. Error	*p*-Value	Odds Ratio	95% CI
Sex(ref = girls)					
Boys	0.466	0.071	<0.001	1.59	1.39–1.83
Age	0.158	0.025	<0.001	1.17	1.11–1.23
Family structure (ref = nuclear family)					
Stem family	−0.155	0.091	0.089	0.86	0.72–1.02
Other family	0.205	0.080	0.011	1.23	1.05–1.44
Mother’s education level (ref = middle school)					
High school	0.194	0.090	0.031	1.21	1.02–1.45
Undergraduate and above	0.071	0.155	0.647	1.07	0.79–1.45
Father’s education level (ref = middle school)					
High school	−0.019	0.090	0.831	0.98	0.82–1.17
Undergraduate and above	−0.137	0.153	0.372	0.87	0.65–1.18
General self-efficacy	−0.025	0.005	<0.001	0.98	0.97–0.98
Academic performance	−0.226	0.035	<0.001	0.80	0.75–0.85
Parents smoke (ref = no parents smoking)					
Only mother	0.460	0.243	0.059	1.58	0.98–2.55
Only father	0.412 ***	0.074	<0.001	1.51	1.31–1.74
Both parents	0.836 ***	0.137	<0.001	2.31	1.77–3.02
Peer smoking (ref = no)					
Yes	1.073 ***	0.081	<0.001	2.92	2.49–3.43

Note: The model controlled for sex, age, mother’s education level, father’s education level, and general self-efficacy. *** *p* < 0.001.

**Table 3 ijerph-20-01048-t003:** Stratified logistic regression models stratified by peer smoking and parental smoking of academic performance on intention to smoke among non-smoking students.

	Academic Performance			
	Estimate	Std. Error	Odds Ratio	95% CI
No-peer smoking	−0.274 ***	0.041	0.76	0.70–0.82
Yes-peer smoking	−0.183 **	0.065	0.83	0.73–0.95
No parents smoking	−0.326 ***	0.056	0.72	0.65–0.81
Only mother smoking	−0.046	0.235	0.96	0.60–1.51
Only father smoking	−0.227 ***	0.047	0.80	0.73–0.87
Both parents’ smoking	−0.021	0.122	0.98	0.77–1.24

Note: All models controlled for sex, age, mother’s education level, father’s education level, and general self-efficacy. ** *p* < 0.01; *** *p* < 0.001.

**Table 4 ijerph-20-01048-t004:** Logistic models examining interactions on intention to smoke among non-smoking students.

Variables	Estimate	Std. Error	Odds Ratio	95% CI
**Model 1** **:**				
Academic performance	−0.395 ***	0.038	0.67	0.63–0.73
Yes-peer smoking	1.358 ***	0.078	3.89	3.34–4.53
Academic performance × Yes-peer smoking	0.188 **	0.070	1.21	1.05–1.38
**Model 2** **:**				
Academic performance	−0.439 ***	0.051	0.64	0.58–0.71
Only mother smoking	0.886 ***	0.244	2.42	1.50–3.91
Only father smoking	0.472 ***	0.072	1.60	1.39–1.85
Both parents smoking	1.140 ***	0.132	3.13	2.42–4.05
Academic performance × Only mother smoking	0.415	0.218	1.52	0.99–2.32
Academic performance × Only father smoking	0.082	0.068	1.09	0.95–1.24
Academic performance × Both parents’ smoking	0.330 **	0.121	1.39	1.10–1.76

Note: ** *p* < 0.01; *** *p* < 0.001.

## Data Availability

The anonymized data are available from the author upon reasonable request.

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
