# Peer review of "Academic Performance and Peer or Parental Tobacco Use among Non-Smoking Adolescents: Influence of Smoking Interactions on Intention to Smoke"

_ijerph, 2023, doi:10.3390/ijerph20021048_

Round 1

Reviewer 1 Report (New Reviewer)

This study aims to examine the to examine the interaction between academic performance and parents/peer tobacco use on adolescents' intention to smoke, after controlling for sex, age, family structure, parental education level, and self-efficacy variables.

The study was conducted in Changchun city, northeastern China, involving 9394 students aged between 9-16 years, who completed a self-filled questionnaire. Results suggest that peer smoking or smoking by both parents reinforces the association between low academic performance and the intention to smoke among adolescents. This approach based on interactions between relevant variables is a way to approach a contextual perspective in adolescent smoking prevention research and intervention.

In my point of view, the paper is interesting and deserves to be published, but some aspects should be clarified or revised. I will present my comments following the main sections of the paper.

Introduction

Well-structured and well-written introduction, covering the concepts relevant to the study.

(lines 41-51) Paragraph on intention should acknowledge and discuss the “gap” between intention to smoke and smoking behavior.

(lines 64-5) “Low academic achievers are commonly not recognized in the campus environment” … and tend to select and to be selected by peers with the same characteristics (e.g., Vitória et al, 2020, already in the references list).

Methods

The sample of this study is big, and participants were selected using multistage stratified cluster sampling. The questionnaire was self-administered with the supervision of trained teachers and school physicians. Ethical topics were considered. Data analysis was adequate regarding the questions of the study.

Data from this study were collected as part of a cross-sectional study on the prevalence of health risk behaviors in primary and secondary school students in Changchun, Jilin Province, China.

(lines 142-3) Authors wrote: “As a representative Chinese city, Changchun is a typical location suitable for studying tobacco use 142 among the youth.” Please clarify what means “a representative Chinese city”. Does Changchun have characteristics that allowed authors to claim that this city is representative the Chinese population? If so, please justify this position.

Results

The results are in line with the hypotheses: the individual effects of academic performance, peer and family members’ tobacco use were found. Significant interactions of academic performance, and peer/family members’ tobacco use on adolescents' intention to smoke were found.

Still, some observations and suggestions:

(lines 252-7) I would like to have more information on differences reported here. Related with this suggestion, please see the next suggestion regarding table 1.

Table 1

The percentages presented in the columns “not intend to smoke” and “intend to smoke” are useless. I would prefer to see percentages by the total of rows than by the total of columns. For example, for sex, it is more useful to know that 91,2% of girls do not intend to smoke comparing with 85% of boys who do not intend to smoke. The same for age… and etc.

Discussion

(lines 339-340) “In recent years” is repeated … “In recent years, globally school smoking ban policies have been implemented in recent years (Pfeifer et al., 2020).”

Limitations

1.

The survey was conducted at schools, and, add also, the questionnaire was self-administrated and answers were not biologically validated… so, participants may have withheld some information… 

2.

To acknowledge the cultural / geographical differences (at least regarding effects of school policies and health education practices on adolescents' perceptions and intentions to smoke).

Future research should investigate the relation between intention to smoke and smoking behavior in this specific population.

Author Response

Response to Reviewer 1 Comments

Point 1:

Introduction:

(lines 41-51) Paragraph on intention should acknowledge and discuss the “gap” between intention to smoke and smoking behavior.

Response 1: Thank the reviewer for the kind reminding. We have add the discussions about the “gap” between intention to smoke and smoking behavior.

The statement are as follows:

Although there is a gap between the number of adolescents who intend to smoke and the prevalence of smoking in matched ages[11], the intention to smoke among adolescents is closely linked to adult smoking, which has been examined in a variety of studies[8, 12-14]. (lines 46-49, page 2).

Point 2:

Methods:

(lines 142-3) Authors wrote: “As a representative Chinese city, Changchun is a typical location suitable for studying tobacco use 142 among the youth.” Please clarify what means “a representative Chinese city”. Does Changchun have characteristics that allowed authors to claim that this city is representative the Chinese population? If so, please justify this position.

Response 2: Thank the reviewer for the kind reminding. The 2013 China Urban Tobacco Prevalence Survey, organized by the Chinese CDC and the US CDC, showed that Changchun's smoking rate was close to the national level. Therefore, we think Changchun is a typical city as a survey about smoking We have revisived the statement in the Methods section. (lines122-126, page3)

The statement are as follows:

The 2013 China Urban Tobacco Prevalence Survey, organized by the Chinese CDC and the US CDC, showed that the overall smoking rate among urban adults (aged15+) in Changchun was 23.5%, with 43.2% for men and 3.8% for women[51], which is similar to national level. Thus, Changchun is a typical city for studying tobacco use among adolescents. (lines122-126, page3)

Point 3:

Results:

Table 1The percentages presented in the columns “not intend to smoke” and “intend to smoke” are useless. I would prefer to see percentages by the total of rows than by the total of columns. For example, for sex, it is more useful to know that 91,2% of girls do not intend to smoke comparing with 85% of boys who do not intend to smoke. The same for age… and etc.

Response 3: Thank the reviewer for the kind suggestion. We have revised Table 1 based on your suggestions.

The new table is as follows:

Table 1. Participant characteristics among students who are non-smokers

Variables

Not intend to smoke

Intend to smoke

Total

c2

p-Value

n (%)

n (%)

n

Sex

girls

4218(91.20)

407(8.80)

4625

85.100

<0.001

boys

4055(85.03)

714(14.97)

4769

Age

~10

1018(92.63)

81(7.37)

1099

152.448

<0.001

~12

3505(91.37)

331(8.63)

3836

~14

3248(85.32)

559(14.68)

3807

³15

362(76.37)

112(23.63)

474

Mother's education level

middle school and below

3953(87.32)

574(12.68)

4527

23.669

<0.001

high school

3053(87.65)

430(12.35)

3483

undergraduate and above

1234(92.09)

106(7.91)

1340

Father's education level

middle school and below

3770(86.81)

573(13.19)

4343

28.021

<0.001

high school

3167(88.27)

421(11.73)

3588

undergraduate and above

1296(92.05)

112(7.95)

1408

Family structure

nuclear family

4292(88.35)

566(11.65)

4858

39.118

<0.001

stem family

2058(90.7)

211(9.3)

2269

other family

1903(84.73)

343(15.27)

2246

General self-efficacy

<20

1333(82.44)

284(17.56)

1617

100.525

<0.001

20-30

4801(87.72)

672(12.28)

5473

>30

2133(92.9)

163(7.1)

2296

Academic performance

poorest

546(79.13)

144(20.87)

690

158.305

<0.001

lower 20%

1552(83.08)

316(16.92)

1868

middle 20%

3193(88.79)

403(11.21)

3596

higher 20%

2247(91.08)

220(8.92)

2467

excellent

711(95.44)

34(4.56)

745

Peer smoking

no

7359(90.63)

761(9.37)

8120

376.446

<0.001

yes

857(71.24)

346(28.76)

1203

Parental smoking

no parents smoke

4073(91.08)

399(8.92)

4472

111.128

<0.001

just mother smoke

114(80.85)

27(19.15)

141

just father smoke

3712(86.37)

586(13.63)

4298

both parents smoke

341(76.8)

103(23.2)

444

Point 4:

Discussion:

(lines 339-340) “In recent years” is repeated … “In recent years, globally school smoking ban policies have been implemented in recent years (Pfeifer et al., 2020).”

Response 4: Thank you for the reminder. Duplicate words have been deleted. (lines325-326,page10)

After revision:

In recent years, globally school smoking ban policies have been implemented. (lines325-326,page10)

Point 5:

Limitations:

1

The survey was conducted at schools, and, add also, the questionnaire was self-administrated and answers were not biologically validated… so, participants may have withheld some information.

2

To acknowledge the cultural / geographical differences (at least regarding effects of school policies and health education practices on adolescents' perceptions and intentions to smoke).

Future research should investigate the relation between intention to smoke and smoking behavior in this specific population.

Response 6: Thank the reviewer for the kind reminding. We have added relevant discussion in the limitations section as suggested. (lines 408-411 , page11; lines 419-423 , page12)

The statement are as follows:

Second, considering that the questionnaire was self-administrated and answers were not biologically validated, participants may have withheld some information in the survey conducted in schools, such as peer smoking and their own intention to smoke. (lines 408-411 , page11)

Fourth, the sample for this study was from the same city. Therefore, cultural and geographical differences were ignored. Future research should also extend to a comparison of adolescents’ smoking intentions in different cultural contexts and consider investigating the effects of local campus policies and health education on adolescents' smoking perceptions and behaviors. (lines 419-423 , page12)

Reviewer 2 Report (New Reviewer)

-Abstract

- More details in the methodology section in the abstract should be added.

-Please add more details also in the results

Introduction

-It is long; I suggest to cut it especially from page 2 line 84 to page 3 line 112.

Methods

-Did the authors get an ethical approval to perform the study, please clarify and add the number of the ethical approval.

-Intention to Smoke questionnaire should be tested for validation and reliability.

-Although the General Self-Efficacy Scale was validated previously in China, however, the validity for this cohort should be tested.

-Please add more details about the software used in the analysis (R 4.0.3)

Author Response

Response to Reviewer 2 Comments

Point 1:

Abstract

- More details in the methodology section in the abstract should be added.

Response 1: Thank the reviewer for the kind reminding. We have added statement to the methods section of the abstract as suggested. (lines10-16, page1)

The statement are as follows:

Methods: A multi-stage stratified sampling was used to select participants, involving 9394 students aged between 9-16 years in Changchun city, northeastern China. Multiple logistic regression analyses were conducted to examine the individual effect of academic performance and peer/parental smoking behavior. Stratified logistic regressions were conducted to examine the protective effect of academic performance based on peer or parental smoking. Interaction effects of academic performance´peer/parental smoking on adolescents’ intention to smoke were tested. (lines10-16, page1)

Point 2:

Abstract

-Please add more details also in the results

Response 2: Thank the reviewer for the kind reminding. Thank the reviewer for the kind reminding. We have added statement to the results section of the abstract as suggested. (lines16-23, page1)

The statement are as follows:

Results: Of all the non-smoking students sampled, 11.9% intended to smoke within the next five years. The individual effect of academic performance and peer/parental smoking was significant. The protective effect of academic performance on the intention to smoke was significant regardless of whether peers smoked or not. However, the protective effect was not significant among adolescents with only maternal smoking and both parental smoking. The current study found the significant interaction effects of academic performance´peer smoking and academic performance´both parents’ smoking. Students with poor academic performance were more likely to intend to smoke if their peers or both parents smoked. (lines16-23, page1)

Point 3:

Introduction

-It is long; I suggest to cut it especially from page 2 line 84 to page 3 line 112.

Response 3: Thank the reviewer for the kind reminding. Along with the reviewer’s suggestion, we have trimmed the introductoion section from page 2, line 84 to page 3, line 112. However, we believe that this part of the introduction is necessary. It explains the mechanisms by which parental and peer smoking affects adolescents' intention to smoke. Therefore, we still retained the part of introduction.

After revision:

According to the social learning theory[1, 2]15], smoking intentions and behavior can be acquired through a learning process in which imitation and differential reinforcement play key roles[3, 4]. This indicates that the perceptions and behaviors of smoking can be affected by environmental exposures, such as intergenerational influences in the family and peer influences from shared settings[5, 6]. Parental smoking behavior may serve as a model for adolescents and be interpreted by their children as a tacit license for addictive substance use[7]. Prior research discovered that adolescents whose parents smoke are more likely to smoke in the future than children in tobacco-free homes[8]. Moreover, a large body of published research supports the idea that their close friends smoking increases the risk of future smoking among adolescents[9], and young people's initiation of smoking is likely triggered by their best friends[10]. In-school groups are a traceable starting point for the developmental trajectory of adolescents' substance use intentions[11, 12]. Once someone in the group smokes, those involved in tobacco use force the negative norms and values on the group members, thereby changing non-smokers’ perceptions about tobacco products to be consistent with the expectations of the group[13]. (lines83-97, page2)

Point 4:

Methods

-Did the authors get an ethical approval to perform the study, please clarify and add the number of the ethical approval.

Response 4: Thank the reviewer for the kind reminding. We have added the ethical statements in the methods section. (lines 150-151, page4)

The statement are as follows:

The study was approved by the Medical Ethics Committee of the School of Public Health Jilin University. The approval number is 2020-10-16. (lines 150-151, page4)

Point 5:

Methods

-Intention to Smoke questionnaire should be tested for validation and reliability.

Response 5: Thank the reviewer for the kind reminding. The intention to smoke in this cross-sectional study survey was asked through a single question. Therefore, it is hard to conduct a reliability or validity test. However, many studies have used the same method to measure adolescents' intention to smoke. And there are longitudinal studies testing the validity of the questions. There is sufficient evidence support that the question, when applied to nonsmoking adolescents, is a valid predictor of future smoking in adolescents.

However, it is true that adolescents' intention to smoke is not entirely reliable. We have added a brief statement in the Introduction section to illustrate this issue. (lines 46-49, page2)

Related Studies:

Pierce, J. P., Choi, W. S., Gilpin, E. A., Farkas, A. J., & Merritt, R. K. (1996). Validation of susceptibility as a predictor of which adolescents take up smoking in the United States. Health Psychology, 15(5), 355-361. https://doi.org/10.1037/0278-6133.15.5.355

Azagba, S., Baskerville, N. B., & Foley, K. (2017). Susceptibility to cigarette smoking among middle and high school e-cigarette users in Canada. Preventive Medicine, 103, 14-19. https://doi.org/10.1016/j.ypmed.2017.07.017

Moodie, C., MacKintosh, A. M., Brown, A., & Hastings, G. B. (2008). Tobacco marketing awareness on youth smoking susceptibility and perceived prevalence before and after an advertising ban. European Journal of Public Health, 18(5), 484-490. https://doi.org/10.1093/eurpub/ckn016

The statement are as follows:

Although there is a gap between the number of adolescents who intend to smoke and the prevalence of smoking in matched ages[14], the intention to smoke among adolescents is closely linked to adult smoking, which has been examined in a variety of studies[15-18]. (lines 46-49, page2)

Point 6:

Methods

-Although the General Self-Efficacy Scale was validated previously in China, however, the validity for this cohort should be tested.

Response 6: Thank the reviewer for the kind reminding. We conducted a factor analysis and found that the KMO value of the General Self-Efficacy Scale is 0.901(>0.85). Bartlett's spherical test showed a significant statistical result (P<0.001). The Cronbach's alpha of the scale was 0.845. Therefore , we believe that the general self-efficacy scale has a good validity and a high reliability. We have added the revelent statement. (lines 192-196, page4-5)

The statement are as follows:

In our study, the KMO value of the General Self-Efficacy Scale is 0.901(>0.85). Bartlett's spherical test showed a significant statistical result (P<0.001). The Cronbach's alpha of the scale was 0.845. These results indicated a good construct validity and a high reliability of the General Self-Efficacy Scale in this survey. (lines 191-195, page4-5)

Point 7:

Methods

-Please add more details about the software used in the analysis (R 4.0.3)

Response 7: Thank the reviewer for the kind reminding. As suggested, we added statements of the data analysis using R software to create interaction terms in Methods section. (lines 212-214, lines 218-220,page5)

The statement are as follows:

Interaction terms were created by multiplying the peer smoking or parental smoking variables and academic performance. (lines 218-219, page5)

This manuscript is a resubmission of an earlier submission. The following is a list of the peer review reports and author responses from that submission.

Round 1

Reviewer 1 Report

The manuscript titled, “Academic Performance and Peer or Parental Tobacco use among Non-Smoking Adolescents: Interactions on Intention to Smoke,” is a cross-sectional study assessing the relationship between academic performance, peer and parental smoking behaviors, and adolescent intention to smoke. Much of the analysis appears to be mythologically sound; however, the text is a bit confusing at times. I believe that the material presented in the paper is interesting and valid. This manuscript appears to fit within the scope of the International Journal of Environmental Research and Public Health.

Regarding the Introduction:

1.       There appears to be a few guiding sentences that were left in the paper. Please remove extra guiding sentences. Example the first sentence is: “The introduction should briefly place the study in a broad context and highlight why it is important.” This should not be in the manuscript.

2.       Unfortunately, there is quite a bit of incorrect word usage and grammar issues through the paper. This makes it much harder to read and leaves the manuscript a bit confusing at times. This is not a fatal flaw it will just require that either the authors will need to do substantial copy editing and proofing or the journal will need to.

3.       The authors go into grater detail into the background of associations between smoking and academics as well as smoking and peer/parental smoking; however, much less is presented beyond the theoretical rationale for the use of intention to smoke. Can this be expanded to include specific results?

Regarding the Methods:

4.       Given that there was strategy in the randomization and data collection procedures could the authors please provide the survey response rate? Also, of those who did complete the survey were they all 100% complete or were some removed due to incomplete responses.

5.       How were incomplete responses handled in the models? Please explain?

6.       I am confused about the sample, the authors state “To screen out non-smokers, all respondents finished questions about smoking experience and current smoking.” But then say that “Former and current smokers were not included in the study.” Please correct and be clear. If the non-smokers were screened out then there would be only smokers? This does not seem to fit with the rest of the manuscript.

7.       Please provide justification for how the outcome measure was operationalized. Why was it dichotomized? Why is “probably no” considered susceptible? Is this based on prior precedent?

Regarding the Results and Discussion:

8.       Please explicitly provide the comparison groups for categorical variables either in the methods or in the results by saying “compared to.”

9.       In table 3 the authors state that these are “multinomial” models? Is this accurate given that the outcome is dichotomous. Please explain in the methods how the multinomial models were constructed or change the word.

10.   Although I do like the figures to display the interaction effects it would be nice to see actual estimates. Would the authors consider developing stratified models to actually find the estimates? Example: Academic Performance -> Intention (Stratified by Peer Smoking).

Author Response

Response to Reviewer 1 Comments

Point 1: There appears to be a few guiding sentences that were left in the paper. Please remove extra guiding sentences. Example the first sentence is: “The introduction should briefly place the study in a broad context and highlight why it is important.” This should not be in the manuscript.

Response 1: Thank the reviewer for the kind reminding. We have deleted the extra guiding sentences.

Point 2: Unfortunately, there is quite a bit of incorrect word usage and grammar issues through the paper. This makes it much harder to read and leaves the manuscript a bit confusing at times. This is not a fatal flaw it will just require that either the authors will need to do substantial copy editing and proofing or the journal will need to.

Response 2: Thank the reviewer for the kind reminding. Language editing has been performed. We have put the “Editing Certificate” in the attachment.

Point 3: The authors go into grater detail into the background of associations between smoking and academics as well as smoking and peer/parental smoking; however, much less is presented beyond the theoretical rationale for the use of intention to smoke. Can this be expanded to include specific results?

Response 3: Thank the reviewer for the kind reminding. Along with the reviewer’s suggestion, We removed some evidence on smoking behavior and added evidence about the associations between intention to smoke and academic performance as well as smoking and peer/parental smoking to the intraduction. (lines56-58, lines61-63, lines90-93, page2;lines97-98,page2-3)

The statement are as follows:

Previous studies have noted that academic performance and intentions to smoke are highly correlated during adolescence(Hyun Gwon et al., 2017; Kinnunen et al., 2016; Pennanen et al., 2011a; Wellman et al., 2016).(lines56-58)

In China, the evidence from a multiethnic survey found that students in the last 25% and lower than 50% of academic achievement are more likely to have an intention to smoke(Xu et al., 2017). (lines61-63)

A study conducted in Hong Kong found that if parents smoked regularly in front of their children at home, adolescents' intention to smoke would be strengthened by the parental role-modeling process(Mak et al., 2012).  (lines90-93)

A previous study in Europe also showed that the smoking behavior of peers was a significant predictor of adolescent intention to smoke(Vitoria et al., 2009).  (lines97-98,page2-3)

Point 4: Given that there was strategy in the randomization and data collection procedures could the authors please provide the survey response rate? Also, of those who did complete the survey were they all 100% complete or were some removed due to incomplete responses.

Response 4: Thank you for the reminder. About the response rate, as we excluded participants who refused to answer, a total of 9893 questionnaires were collected after distributing 10157 questionnaires and the response rate was 97.4%. Those who had missing dependent variable (intention to smoke) (N=255) and logical contradictions (N=17) in their answers were removed.

We have added the following statement to the section 2.1. Data Collection and Participants (lines 142-146, page 3).

The statement are as follows:

As we excluded participants who refused to answer, a total of 9893 questionnaires were collected after distributing 10157 questionnaires and the response rate was 97.4%. After removing samples with smoking experience (N=227), and had missing dependent variable (intention to smoke) (N=255) and logical contradictions (N=17) in their answers, 9394 participants were selected.  (lines 142-146, page 3).

Point 5: How were incomplete responses handled in the models? Please explain?

Response 5: Thank you for the reminder. In the regressions, the missing values of the independent variables were replaced by multiple imputation. (lines 227-228, page5) The statement on this part has been added in 2.3 Data analysis.

Point 6: I am confused about the sample, the authors state “To screen out non-smokers, all respondents finished questions about smoking experience and current smoking.” But then say that “Former and current smokers were not included in the study.” Please correct and be clear. If the non-smokers were screened out then there would be only smokers? This does not seem to fit with the rest of the manuscript.

Response 6: Thank the reviewer for the kind reminding.The participant of this study were adolescents who had no smoking experience. The statement on this part has been revised. (lines 151-152 , page4)

The statement are as follows:

To screen out smokers, all respondents completed questions about smoking experience and current smoking.  (lines 151-152, page4)

Point 7: Please provide justification for how the outcome measure was operationalized. Why was it dichotomized? Why is “probably no” considered susceptible? Is this based on prior precedent?

Response 7: Thank the reviewer for the kind reminding.The basis for dichotomous was referenced from Pierce et al.'s method of assessing smoking susceptibility. (lines 186-187, page4). We have added relevant references.

Reference:

Pierce, J. P., Choi, W. S., Gilpin, E. A., Farkas, A. J., & Merritt, R. K. (1996). Validation of susceptibility as a predictor of which adolescents take up smoking in the United States. Health Psychology, 15(5), 355-361. https://doi.org/10.1037/0278-6133.15.5.355

Azagba, S., Baskerville, N. B., & Foley, K. (2017). Susceptibility to cigarette smoking among middle and high school e-cigarette users in Canada. Preventive Medicine, 103, 14-19. https://doi.org/10.1016/j.ypmed.2017.07.017

Moodie, C., MacKintosh, A. M., Brown, A., & Hastings, G. B. (2008). Tobacco marketing awareness on youth smoking susceptibility and perceived prevalence before and after an advertising ban. European Journal of Public Health, 18(5), 484-490. https://doi.org/10.1093/eurpub/ckn016

Point 8: Please explicitly provide the comparison groups for categorical variables either in the methods or in the results by saying “compared to.”

Response 8: Thank the reviewer for the kind reminding. we have added the comparison groups for categorical variables in the results. The statement on this part has been revised as follows.(lines 255-261, page 6)

The statement are as follows:

Compared to adolescents without peer smoking, the students with peer smoking (OR = 2.92, 95% CI = 2.49-3.43) were more likely to smoke. Students in smoke family were more likely to smoke than students whose parents did not smoke: (1) just father smoking (OR = 1.51, 95% CI = 1.31-1.74) and (2) both parents smoking (OR = 2.31, 95% CI = 1.77-3.02). In addition, sex (ref=girls), age, family structure (ref=nuclear family) and general self-efficacy were significantly associated with the intention to smoke. (lines 255-261, page 6)

Point 9: In table 3 the authors state that these are “multinomial” models? Is this accurate given that the outcome is dichotomous. Please explain in the methods how the multinomial models were constructed or change the word.

Response 9: Thank the reviewer for the kind reminding. Dichotomous logistic regression was used to test the interaction term. The statement on this part has been revised.(lines 304, page 9)

The statement are as follows:

Table 4. logistic model examining interactions on intention to smoke among nonsmoking student.  (lines 305, page 8)

Point 10: Although I do like the figures to display the interaction effects it would be nice to see actual estimates. Would the authors consider developing stratified models to actually find the estimates? Example: Academic Performance -> Intention (Stratified by Peer Smoking).

Response 10: Along with the reviewer’s suggestion, we developed stratified models to examin the associations of parental smoking, peer smoking and academic performance with intention to smoke. We have added the following content to the 2.3 Data Analysis and Results section. (lines227-228, page5; lines262-267, page6; and Table3)

The statement are as follows:

Subsequently, stratified models was developed to examin the associations of parental smoking, peers’ smoking, and academic performance with intention to smoke.  (lines222-224, page5)

As shown in Table 3, in stratified models, for students without smoking peers, the OR of academic performance was 0.83 (0.78–0.89); with smoking peers, the OR of academic performance was 0.76 (0.73–0.79). Without smoking parents, the OR of academic performance was 0.72 (0.67–0.76); with just the father smoking, the OR of academic performance was 0.80 (0.76–0.84). For those just the mother smokes or both parents smoke, the effect of academic performance is not significant.  (lines264-269, page8)

Table 3. Stratified logistic regression models stratified by peer smoking and parents smoking of academic performance on intention to smoke among non-smoking students.

Academic Performance

Estimate

Std. Error

Odds Ratio

95%CI

No-peer smoking

-0.274***

0.041

0.76

0.70-0.82

YES-peer smoking

-0.183**

0.065

0.83

0.73-0.95

No parents smoking

-0.326***

0.056

0.72

0.65-0.81

Just mother smoking

-0.046

0.235

0.96

0.60-1.51

Just father smoking

-0.227***

0.047

0.80

0.73-0.87

Both parents’ smoking

-0.021

0.122

0.98

0.77-1.24

Note: All models controlled for sex, age, mother's education level, father's education level, general self-efficacy.

* p<0.05; ** p<0.01; *** p<0.001   (lines300-304, page 8-9)

Reviewer 2 Report

Authors provide a cross-sectional study on the prevalence of health risk behaviors of primary and secondary school students in Changchun, Jilin Province, China.

Introduction: As currently written, the entire manuscript contains several grammatical and syntax errors that detracts from the quality of the manuscript. 

Lines 28-29 - remove introductory sentence " The introduction should briefly place the study in a broad context and highlight why it is important".

The authors fail to show the different effects adult smoking has on the region and/or country of interest. In other words, if we know adolescent smoking is low and adult smoking is high in China, what novel question and hypothesis are generated with this study? Who does high prevalence of adult smoking affect China and/or Changchun province specifically? 

Methods: a major pitfall of the study is clearly differentiating between primary/secondary age students, youth, and adolescents. As currently written, it remains unclear how sampling primary and secondary age students is similar or different than the evidence provided from previously published data involving youth and adolescents. 

If the impact of parent/peer smoking is linked to adolescent intention to smoke as adults, why is not plausible that any effect observed during primary or secondary age will change in an age-dependent manner? 

Results: the table descriptions are insufficient. 

What is the average age of the survey participants? 

What would be the average age of the survey participants in 5 years (in reference to survey question)

The data does not reflect the fact that most adolescents do not choose to smoke; yet, the questions in the survey are examining the intention to smoke as adolescents. 

Discussion/Conclusion: It remains unclear how the authors clearly delineated between 'peer' and 'parent'. Further discussion is needed to explore the complexity of age, the concept of intention in an age-dependent manner, and the role of education. 

As currently written it remains unclear why academic achievement was selected. 

Although the study provides an important data set regarding smoking prevalence, the manuscript needs significant improvement. 

Author Response

Response to Reviewer 2 Comments

Point 1: Introduction: As currently written, the entire manuscript contains several grammatical and syntax errors that detracts from the quality of the manuscript. 

Response 1: Thank the reviewer for the kind reminding. Language editing has been performed. We have put the “Editing Certificate” in the attachment.

Point 2: Lines 28-29 - remove introductory sentence " The introduction should briefly place the study in a broad context and highlight why it is important".

Response 2: Thank the reviewer for the kind reminding. We have delete the extra guiding sentences.

Point 3: The authors fail to show the different effects adult smoking has on the region and/or country of interest. In other words, if we know adolescent smoking is low and adult smoking is high in China, what novel question and hypothesis are generated with this study? Who does high prevalence of adult smoking affect China and/or Changchun province specifically? 

Response 3: Thank the reviewer for the kind reminding. Along with the reviewer’s suggestion, we added the adult smoking prevalence in Iran and Singapore. The comparison revealed that the adult smoking rates in China are much higher than Iran and Singapore. So we speculate that behavioral exposures resulting from high social smoking rates in Chinese society may motivate nonsmoking adolescents to become smokers as adults. We have revised the statement to the Introduction section as follows. (lines 30-42, page1)

The statement are as follows:

However, the adult smoking rate in China is one of the highest in the world at 26. 6%(Zhang et al., 2022), compared to 12% in Iran (Moosazadeh et al., 2015) and only 16% in Singapore(Shahwan et al., 2019). As smoking prevalence among adolescents is low but adult smoking prevalence is high in China, we speculate that behavioral exposures resulting from high social smoking rates in Chinese society may motivate nonsmoking adolescents to become smokers as adults(Alesci et al., 2003). That means former nonsmoking adolescents are influenced by smokers in their families or communities to gradually become smokers(Klein et al., 2012; Rachiotis et al., 2010). Specifically, the process can be long and adolescents may not decide to smoke immediately due to practical constraints but rather try to smoke in the future after developing an intention to smoke(Wakefield et al., 2004).  (lines 30-40, page1)

Point 4: A major pitfall of the study is clearly differentiating between primary/secondary age students, youth, and adolescents. As currently written, it remains unclear how sampling primary and secondary age students is similar or different than the evidence provided from previously published data involving youth and adolescents. 

Response 4: Thank the reviewer for the kind reminding. After the revisions, this paper no longer explicitly distinguishes between elementary and middle school students. We have run a new chi-square test for intention to smoke and have recreated Table 1. Age was used to replace school type to describe intention to smoke(in table 1). A comparison with other published studies can be seen in lines 321-324, page10

The statement are as follows:

In the current study, 11.9% of nonsmoking adolescents aged 9-16 years in Changchun reported being likely to smoke within 5 years, which was slightly higher than the 9.7% of Chinese middle school students aged 13-15 years old found in another study conducted in 2015(Xu et al., 2017) (lines 321-324, page10)

A part of Table 1

Variables

Not intend to smoke

Intend to smoke

Total

2

P-Value

N (%)

N (%)

N (%)

Age

~10

1018(12.5)

81(7.5)

1099(11.9)

152.448

<0.001

~12

3505(43.1)

331(30.6)

3836(41.7)

~14

3248(39.9)

559(51.6)

3807(41.3)

15

362(4.5)

112(10.3)

474(5.1)

Point 5: If the impact of parent/peer smoking is linked to adolescent intention to smoke as adults, why is not plausible that any effect observed during primary or secondary age will change in an age-dependent manner?

Response 5: Thank the reviewer for the kind reminding.The complex role of age on intention to smoke is possible to exist. However, about this question, we didn't take into account carefully, thus the current investigation has limited coverage of relevant content. While it is true that influences from school and family may be effective in the long term, adolescence is a special time when behavioral and attitudinal changes are susceptible to environmental influences. We have added some satemnet in the limitations section to account for this issue as follows (lines 399-410, page 12)

The statement are as follows:

Third, this study focused on adolescents aged 9-16 years, although the influences from school and family have been shown to be effective over time(Spoth et al., 2005), adolescence is a special period with rapid cognitive development, and many underlying psychological characteristics and attitudes toward addictive substances may be age-related. However, the current investigation has limited coverage of relevant content. In addition, age is a risk factor for adolescent intention to smoke in the current study. Nevertheless, as numerous tobacco control policies are implemented and health education becomes more widespread, adolescents' increasing age and cognitive function may lead them to resist tobacco exposure in the home, school, and community. Future research can attempt to identify and explain age-related changes of adolescent samples and also consider investigate the effects of campus policies and health education on adolescents' perceptions and intentions to smoke.  (lines 398-409, page 12)

Point 6: The table descriptions are insufficient. 

Response 6: Thank the reviewer for the kind reminding. We have re-run the chi-square test for intention to smoke and have recreated Table 1. We have also modified the description of Table1 as follows. (lines 233-250, page 5-6)

The statement are as follows:

Overall, a total of 9394 nonsmoking students aged 9-16 participated in the study. The mean age (and SD) of the non-smoking students was 12.32 (1.5) years. 8273 (88%) of the nonsmoking students reported definitely not smoking within the next five years, 797 (8.5%) reported probably not, 288 (3.1%) reported probably yes, and 36 (0.4%) reported definitely yes.

Of the participants, 50.8% were boys;41.7% were 10(not include)-12 years old; 48.4% reported their mothers’ highest education was middle school and below; 46.5% reported their fathers’ highest education was middle school and below; 51.8% lived in a nuclear family; 58.3% had general self-efficacy scores between 20 to 30.; 38.4% reported their academic performance in middle 20%. In general, 19.8% reported that peers smoke; 47.8 reported that neither parent in family smoked, 1.5 reported just mother smoke, 45.9% reported just father smoke, 1.9% reported both parents smoke. Statistically significant differences were observed in sex(c2=85.1, P<0.001), age(c2=85.1, P<0.001), mother's education level(c2=23.669, P<0.001), father's education level(c2=28.021, P<0.001), family structure(c2=39.118, P<0.001), general self-efficacy(c2=100.525, P<0.001), academic performance(c2=158.305, P<0.001), peer smoke(c2=376.446, P<0.001) and parental smoke(c2=111.128, P<0.001) between students who intend to smoke and not intend to smoke (Table 1).  (lines 233-250, page 5-6)

Table 1. Participant characteristics among students who are nonsmokers.

Variables

Not intend to smoke

Intend to smoke

Total

2

P-Value

N (%)

N (%)

N (%)

Sex

girls

4218(51.0)

407(36.3)

4625(49.2)

85.100

<0.001

boys

4055(49.0)

714(63.7)

4769(50.8)

Age

~10

1018(12.5)

81(7.5)

1099(11.9)

152.448

<0.001

~12

3505(43.1)

331(30.6)

3836(41.7)

~14

3248(39.9)

559(51.6)

3807(41.3)

15

362(4.5)

112(10.3)

474(5.1)

Mother's education level

middle school and below

3953(48.0)

574(51.7)

4527(48.4)

23.669

<0.001

high school

3053(37.0)

430(38.7)

3483(37.3)

undergraduate and above

1234(15.0)

106(9.6)

1340(14.3)

Father's education level

middle school and below

3770(45.8)

573(51.8)

4343(46.5)

28.021

<0.001

high school

3167(38.5)

421(38.1)

3588(38.4)

undergraduate and above

1296(15.7)

112(10.1)

1408(15.1)

Family structure

nuclear family

4292(52.0)

566(50.5)

4858(51.8)

39.118

<0.001

stem family

2058(24.9)

211(18.8)

2269(24.2)

other family

1903(23.1)

343(30.7)

2246(24.0)

General self-efficacy

<20

1333(16.1)

284(25.4)

1617(17.2)

100.525

<0.001

20-30

4801(58.1)

672(60.1)

5473(58.3)

>30

2133(25.8)

163(14.5)

2296(24.5)

Academic performance

poorest

546(6.6)

144(12.9)

690(7.3)

158.305

<0.001

lower 20%

1552(18.8)

316(28.3)

1868(19.9)

middle 20%

3193(38.7)

403(36.1)

3596(38.4)

higher 20%

2247(27.2)

220(19.7)

2467(26.4)

excellent

711(8.7)

34(3.0)

745(9.0)

Peer smoke

no

7359(89.6)

761(68.7)

8120(87.1)

376.446

<0.001

yes

857(10.4)

346(31.3)

1203(12.9)

Parental smoke

no parents smoke

4073(49.4)

399(35.8)

4472(47.8)

111.128

<0.001

just mother smoke

114(1.4)

27(2.4)

141(1.5)

just father smoke

3712(45.0)

586(52.6)

4298(46.0)

both parents smoke

341(4.2)

103(9.2)

444(4.7)

Point 7: What is the average age of the survey participants? 

Response 7: Thank the reviewer for the kind reminding. The mean age (and SD) of the nonsmoking students was 12.32 (1.5) years. (lines 233-234, page5) We have added a description of the average age to the results section.

Point 8: What would be the average age of the survey participants in 5 years (in reference to survey question)

Response 8: The average age of the survey participants in 5 years is 17.32

Point 9: The data does not reflect the fact that most adolescents do not choose to smoke; yet, the questions in the survey are examining the intention to smoke as adolescents. 

Response 9: Thank the reviewer for the kind reminding.We have examined the intention to smoke among adolescents and recreat the table 1. New table1 shows most adolescents do not choose to smoke. In the current study, 11.9% of nonsmoking adolescents aged 9-16 years in Changchun reported being likely to smoke within 5 years. (lines 319-320,page10)

Point 10: It remains unclear how the authors clearly delineated between 'peer' and 'parent'. Further discussion is needed to explore the complexity of age, the concept of intention in an age-dependent manner, and the role of education.

Response 10: Thank the reviewer for the kind reminding. In the discussion section, the roles of peers and parents have been discussed in their respective segments. We have modified the interaction results of peer smoking*academic performance and parental smoking*academic performance.(355-358,page11; 366-369,page11) About the role of education, along with the reviewer’s suggestion, we have added some statement in the Disscussion section(337-339,page 11), but we were unable to explain the complexity of age and education due to the limitations of the survey content. Therefore, statements was also added to the limitations section. (same as response 5) (lines 398-409, page 12).

the relevant discussion about education has been added to the Dission section as follows:

School health education is also an important part of tobacco control for adolescents, thus health education about smoking prevention is necessary in the school curriculum(Rakete et al., 2010; Xiao et al., 2018)  (337-339,page 10-11)

Point 11: As currently written it remains unclear why academic achievement was selected. 

Response 11: School and home are the basic environment for adolescent to grow up. Academic achievement reflects a youth's school experience and accomplishments. And there are many studies finding the association between adolescents' intention to smoke and academic achievement. Moreover, it is entirely possible that there is a combined effect of school and family. Therefore, we chose academic achievement. we have added the following statement to the Introduction and Disscussion section.(lines 53-58,page1; lines 344-350, page 11)

The statement are as follows:

School is the primary environment in which adolescents grow up and where they acquire the necessary knowledge and skills to succeed in life. Studies have placed special emphasis on the impact of the school environment on children's behaviors and attitudes(Beck & Reilly, 2017; Bonell et al., 2013). Previous studies have noted that academic performance and intentions to smoke are highly correlated during adolescence(Hyun Gwon et al., 2017; Kinnunen et al., 2016; Pennanen et al., 2011a; Wellman et al., 2016).  (lines 53-58,page2)

Academic performance is an important indicator of a campus success, not only related to the adolescent's own sense of accomplishment, but also to the social and emotional support they received(Elstad, 2010). High academic achievers will receive more recognition from the school environment and family, while low achievers have to endure neglect from teachers and peers, and pressure from parents. A study showed that students who lacked social support had a higher intention to smoke(Roberts et al., 2016).  (lines 345-351, page 11)

Round 2

Reviewer 2 Report

Please make a final check of grammar and spelling. 

No further comments or edits